# Image Classification and Automated Machine Learning to Classify Lung Pathologies in Deceased Feedlot Cattle

**DOI:** 10.3390/vetsci10020113

**Published:** 2023-02-03

**Authors:** Eduarda M. Bortoluzzi, Paige H. Schmidt, Rachel E. Brown, Makenna Jensen, Madeline R. Mancke, Robert L. Larson, Phillip A. Lancaster, Brad J. White

**Affiliations:** Beef Cattle Institute, Kansas State University, Manhattan, KS 66506, USA

**Keywords:** acute interstitial pneumonia, bovine respiratory disease, bronchopneumonia with an interstitial pattern, feedlot necropsy, image classification

## Abstract

**Simple Summary:**

Respiratory syndromes are the main cause of ill and deceased animals in the feedlot industry. The correct diagnostic of lung lesions is important to prevent and adapt managements strategies within feedyards. The necropsy of deceased animals is veterinarians’ main tool for postmortem diagnoses; however, it is prone to time and location constraints. Necropsy image analysis can be used to overcome these challenges. Image classification models using machine learning were developed to determine respiratory syndromes’ diagnostic accuracy in right lateral necropsied feedlot cattle lungs. Models performed better at classifying bovine respiratory disease and bronchopneumonia with an interstitial pattern compared to acute interstitial pneumonia. Models developed still require fine-tuning; however, they present potential to assist veterinarians in diagnosing lung diseases during field necropsies.

**Abstract:**

Bovine respiratory disease (BRD) and acute interstitial pneumonia (AIP) are the main reported respiratory syndromes (RSs) causing significant morbidity and mortality in feedlot cattle. Recently, bronchopneumonia with an interstitial pattern (BIP) was described as a concerning emerging feedlot lung disease. Necropsies are imperative to assist lung disease diagnosis and pinpoint feedlot management sectors that require improvement. However, necropsies can be logistically challenging due to location and veterinarians’ time constraints. Technology advances allow image collection for veterinarians’ asynchronous evaluation, thereby reducing challenges. This study’s goal was to develop image classification models using machine learning to determine RS diagnostic accuracy in right lateral necropsied feedlot cattle lungs. Unaltered and cropped lung images were labeled using gross and histopathology diagnoses generating four datasets: unaltered lung images labeled with gross diagnoses, unaltered lung images labeled with histopathological diagnoses, cropped images labeled with gross diagnoses, and cropped images labeled with histopathological diagnoses. Datasets were exported to create image classification models, and a best trial was selected for each model based on accuracy. Gross diagnoses accuracies ranged from 39 to 41% for unaltered and cropped images. Labeling images with histopathology diagnoses did not improve average accuracies; 34–38% for unaltered and cropped images. Moderately high sensitivities were attained for BIP (60–100%) and BRD (20–69%) compared to AIP (0–23%). The models developed still require fine-tuning; however, they are the first step towards assisting veterinarians’ lung diseases diagnostics in field necropsies.

## 1. Introduction

Respiratory diseases and disorders are reported as the main causes of morbidity and mortality in feedlot cattle [1,2]. Economic losses related to bovine respiratory disease (BRD) and acute interstitial pneumonia (AIP) are economically significant and negatively impact the beef cattle industry [2,3,4,5]. Research mainly focuses on the two major reported respiratory disease syndromes: BRD and AIP. Differentiation between respiratory diseases can be challenging when only clinical presentations are considered, since symptoms are not exclusive of just one disease. Respiratory distress signs such as neck extension, open-mouth breathing, and nasal and ocular discharge are commonly observed when the respiratory tract is affected [6,7,8]. Visual signs of respiratory distress have moderate accuracy due to variability in subjective evaluation and biological variation in disease presentation [9]. Recently, bronchopneumonia with an interstitial pattern (BIP) diagnosis was described as an emerging lung disease in feedlot cattle [10]. The report of BIP cases raised the questions of respiratory diseases possibly being inaccurately diagnosed and only classified as AIP and BRD.

Necropsies can provide important information regarding the cause of death and the accuracy of animal health surveillance and indicate management areas that can be improved [11,12]. Differences in pathomorphological presentations of BRD, AIP, and BIP can be identified and assist the diagnosis during necropsies. Accurate diagnosis assists in creating a specific and appropriate treatment and prevention program for different respiratory disease syndromes. The gross presentation of BRD can be dependent on the pathogen associated with the disease, varying from suppurative bronchopneumonia, fibrinous pneumonia or fibrous pleuropneumonia, and caseonecrotic bronchopneumonia [13]. Lesions are commonly observed as expanded and firm consolidated areas in the cranioventral lobes, and possibly presenting interlobular septa distention, edema, fibrinous pleuritis, and alveolar emphysema [13,14]. Pathology lung features of AIP include edematous and emphysematous firm lung lobes, which are non-collapsible and enlarged with an acute diffuse distribution of interstitial pattern present [13,15,16]. Visually, lesions are commonly notable on caudodorsal lobes, presenting a mosaic or “checkerboard” pattern [17]. The gross pathological findings of BIP were recently first described [18]. BIP lungs present features of both BRD and AIP and were described as consolidated and pneumonic cranioventral lobes with a sharp demarcation line separating from the interstitial pattern of the caudodorsal lungs.

Despite different gross pathological presentations of these diseases, necropsies by trained personnel to differentiate among syndromes can be logistically challenging due to the location and time constraints of veterinarians. Computer imaging technology has been used since the beginning of the 20th century, where videos and images were collected by trained necropsy technicians located within the feedlots and sent to remote attending veterinarians to establish the cause of death and generate a necropsy report [19]. The constant advances in image quality, digital storage, and overall technology improvement means that images can now be used with machine learning techniques to assist field diagnosis. Artificial intelligence has been applied to image analysis mainly using radiographs and MRI in veterinary medicine, but no reports were found regarding application to necropsy images [20,21,22,23].

To the authors’ knowledge, beef cattle lung images obtained at necropsy have not been used to classify lesions. This study aims to develop image classification models for images of the right lateral orientation of feedlot cattle lungs collected during field necropsy and to determine the accuracy of these models for diagnosing common respiratory syndromes. The goal is to use the artificial-intelligence-derived models to assist in respiratory disease diagnosis and differentiation between AIP, BIP, BRD, and other diagnoses during feedlot cattle field necropsy.

## 2. Materials and Methods

No live animals were utilized in this project; all photographs were collected from animals deceased from natural causes within the feedyard operation. The Kansas State University Institutional Animal Care and Use Committee was contacted, and an IACUC application was deemed not necessary for this project.

### 2.1. Images

Right lateral lung images were collected during feedlot cattle field necropsies from 6 feedyards in the High Plains region of United States between June and August 2022. Necropsies were performed by a team of 4 veterinary technicians with the supervision of a veterinarian. A total of 398 images were recorded, using a Canon EOS 4000D DSLR Camera with an EF-S 18–55 mm lens (Canon Inc., Tokyo, Japan), and uploaded to a storage account container within the Microsoft Azure platform. The Microsoft Azure machine learning studio was used to label the images according to gross, and when available, histopathology diagnoses for each case. The possible gross diagnoses (Figure 1) of this project were: acute interstitial pneumonia (AIP), bronchopneumonia with an interstitial pattern (BIP), bovine respiratory disease (BRD), and other lung diagnoses that were not AIP, BIP, or BRD (undifferentiated, i.e., embolic pneumonia, healthy lungs, and other diagnoses).

The necropsy technicians and a veterinarian conjointly defined the gross diagnoses for each lung image during the necropsy procedure. Histopathological samples were taken on a subset of cattle from each feedyard each week, including cattle deceased from diverse gross diagnoses. The histopathology diagnoses were obtained by the histological analysis of four lung samples (left cranioventral lobe; right cranioventral lobe; left caudal dorsal lobe; right caudal dorsal lobe). The sample collection for histopathology was developed with the intent to capture major lung sections commonly affected by pulmonary pathologies (cranioventral and caudodorsal lobes). The four possible histology diagnoses were the following: acute interstitial pneumonia (AIP), bronchopneumonia with an interstitial pattern (BIP), bovine respiratory disease (BRD), and other diagnoses (Undifferentiated). The overall calf histopathological diagnoses were considered BRD if only BRD was identified in all four lung samples, AIP if only AIP was identified in all four lung samples, BIP if both AIP and BRD were identified in any combination of the four lung samples, and undifferentiated if none of the above were identified in any of the four lung samples.

### 2.2. Data Labeling

Data labeling was carried out following a sequence of steps (Figure 2) on the Azure Machine Learning Studio (Microsoft Corporation, Redmon, WA, USA). In the first step, unaltered lung images were labeled using the image classification (multi-class) function, and the data were exported to be modeled using the automated ML feature. In the second step, unaltered images were labeled using object identification (bounding box) to locate and select the lung in the image. Then, data were exported and fed to the automated ML feature to model lung image classification. The model used to train object detection was Yolov5, and its best run was utilized to crop the training images using the notebook feature of Azure. The cropped images were utilized to minimize the background noise. These images were uploaded to the storage account and fed back into Azure Machine Learning to be labeled using image classification (multi-class).

A total of four image classification projects were created (Figure 3), including unaltered lung images labeled with gross diagnoses, unaltered lung images labeled with histopathological diagnoses, cropped images labeled with gross diagnoses, and cropped images labeled with histopathological diagnoses. Data originating from these four projects were exported to create an image classification model using the automated ML feature.

### 2.3. Modeling

Exported tabular data from image classification were selected to be used in the automated ML training studio of Microsoft Azure. The task type was automatically selected as image classification, and the target column was set as label (string). A machine learning compute GPU (1 x NVIDIA Tesla k80) of 6 cores, 56 GB Ram, and a 380 GB disk was used to run the classification algorithms. Four model algorithms for the image classification task were selected: MobileNet (light-weighted model for mobile applications), SE-ResNeXt (squeeze and excitation networks), ResNet50 (residual networks), and ViTl16r224 (vision transformer networks). A model agonistic hyperparameter was fitted to the algorithms, controlling the number of training epochs (number_of_epochs). In random sampling, 20 iterations with 4 concurrent iterations were used as tuning settings. Data sets were split 80% for training and 20% for testing. The results reported were selected from the best trial based on the accuracy of image classification models computed using automated ML.

## 3. Results

### 3.1. Descriptive Statistics

Images taken encompass a cohort of 398 necropsied cattle with an estimated weight at time of death of 445 ± 93 kg and 90 ± 52 days on feed, representing mainly beef cross breeds (*n* = 372), beef–dairy cross breeds (*n* = 23), and dairy breeds (*n* = 3). Furthermore, four different data sets were created from unaltered and cropped images and paired with the gross and histopathology diagnoses (Figure 3). The number of images per set and their labels are represented in Table 1. A total of 398 right lateral cattle lung images were labeled in the unaltered images and gross diagnoses set. From those, 35.4% of the images were labeled as BIP, 11% were labeled as AIP, 37.2% were labeled as BRD, and 16.4% were labeled as undifferentiated. From the 398 unaltered lung images, the cropped model used only 80% of images from the training set, totaling 318 cropped images. These cropped images were labeled according to their histopathology diagnoses. As a result, 37.7% of the images were labeled as BIP, 12.9% were labeled as AIP, 35.5% were labeled as BRD, and 13.9% were labeled as undifferentiated. One hundred and sixty-seven right lateral cattle lung images were labeled using the unaltered images and histopathology diagnoses. From those, 40.1% of the images were labeled as BIP, 12.6% as AIP, 34.1% as BRD, and 13.2% as undifferentiated. The number of images labeled in this set corresponds to samples submitted for histopathology analysis. For the cropped images and histopathology diagnoses set, eighteen images were not available to label due to the reduction in images resulting from the crop model. Thus, 149 images were labeled according to their histology diagnosis. From those, 40.9% of images were labeled as BIP, 10.7% were labeled as AIP, 37.9% were labeled as BRD, and 11.4% were labeled as undifferentiated.

### 3.2. Image Classification Models Performance

#### 3.2.1. Best Model Accuracies

The best trial algorithm for each data set was selected based on the accuracy metric within Azure. The best algorithms were MobileNet, SE-ResNeXt, SE-ResNeXt, and ViTl16r224 for unaltered images gross diagnoses, unaltered images histopathology diagnoses, cropped images gross diagnoses, and cropped images histopathology diagnoses, respectively. The distributions of best model accuracies were calculated within the image classification model for each dataset, and the variation represents the epoch level within training (i.e., when the entire dataset passes forwards and backwards through the neural network) (Figure 4). The average accuracies were 39% and 41% for unaltered and cropped images, respectively, labeled with gross diagnoses. The average accuracies were 34% and 38% for unaltered and cropped images, respectively, labeled with histopathology diagnoses.

#### 3.2.2. Area under the Curve

The area under the curve (AUC) under the receiver operating characteristic curve (ROC) representing each label within each dataset is represented in Table 2. The highest AUC was 0.91 for images labeled undifferentiated for the predictive model developed for cropped images and gross diagnosis. The lowest AUC was 0.44 for images labeled AIP for the predictive model developed for unaltered images and histopathology diagnosis.

#### 3.2.3. Sensitivity, Specificity, and Accuracy

Sensitivity and specificity had a wide range within each dataset (Table 2). Throughout, AIP labels had the lowest sensitivities and highest specificities. BIP images were correctly labeled as true positives (>60% sensitivity) more often compared to AIP and undifferentiated labels. BRD had an acceptable sensitivity (≥60%) for unaltered images with gross diagnoses and cropped images with histopathology diagnoses compared to the other two models (≤31% sensitivity). Class accuracies ranged from 86% to 38%. AIP and undifferentiated classes had greater accuracies compared to BIP and BRD.

#### 3.2.4. Positive and Negative Predictive Values

Positive predictive values (PPVs) ranged from 0 to 75% for the AIP class, 42 to 50% for the BIP class, 40 to 100% for the BRD class, and 0 to 100% for the undifferentiated class (Table 2). Negative predictive values (NPVs) ranged from 76 to 89% for the AIP class, 75 to 100% for the BIP class, 64 to 75% for the BRD class, and 82 to 94% for the undifferentiated class.

## 4. Discussion

Respiratory diseases are the main causes of morbidity and mortality in feedlot cattle, causing a significant impact on animal health and negatively affecting animal production. Postmortem necropsies are the principal tool used by veterinarians to grossly differentiate bovine respiratory disease (BRD), bronchopneumonia with an interstitial pattern (BIP), and acute interstitial pneumonia (AIP). This differentiation is important to pinpoint management strategies that can be implemented to prevent future morbidities and mortalities. With advances in technology, digital storage capabilities, and accessibility, images are being used as a diagnostic tool when veterinarians are not present at the time of animal necropsy, especially due to time and location constraints [19]. More recently, image classification and automated machine learning have been implemented in human and veterinary medicine as auxiliary diagnostic tools [20,21,24,25]. Similar technology was applied to identify postmortem organ images in humans, and it is expected to assist human disease diagnosis in the near future [26]. We evaluated the use of unaltered and cropped right lateral lung images collected during feedlot necropsies and paired them with gross and histopathology labels, creating four datasets. Image classification models were then developed to classify and label lung images as presenting respiratory lesions (diseases) affecting feedlot cattle: AIP, BIP, BRD, and from other undifferentiated lung diagnoses that were not AIP, BIP, or BRD.

In the best models using unaltered and cropped images with gross diagnoses, BIP and BRD presented greater sensitivities compared to AIP and undifferentiated. Cropping the images improved the BIP and decreased BRD sensitivities. However, both models using gross diagnoses incorrectly classified some of the BRD and AIP lung images as BIP class, affecting the specificity of the models. Misclassifications of BIP lung images as the BRD class were also seen in the best model for unaltered images with gross diagnoses, but far less present in the best model for cropped images with gross diagnoses. The misclassification of AIP and BRD lung images were not surprising since grossly BIP lesions encompass bronchopneumonia in the cranioventral portion of the lungs with interstitial pneumonia in the caudal dorsal portion of the lung [18]. These lesions are compatible with gross lesions found separately in cases of AIP and BRD.

The AIP class had the lowest sensitivities for the models using gross diagnoses. Nevertheless, specificity was adequate, yielding good NPV and accuracy outcomes. Grossly AIP is characterized by firm and heavy lungs with an acute diffuse distribution of interstitial pattern [13,17]. The pneumonic lobules appear pale and interposed with dark consolidated or congested lobules, giving the appearance of a checkerboard pattern, and these alterations are visible when observing the images. The AIP gross diagnosis has a tactile component where lung lobules feel rubbery, and exudate is present from cut surfaces; therefore, AIP may not be accurately identified visually alone (i.e., without palpation and manipulation) [13]. The tactile component of AIP might be the missing piece from the image classification models created in this study since only pixels and vectors are used to classify and categorize the lung images. The undifferentiated lung images had 0% sensitivity in the best model for unaltered images with gross diagnoses, and good sensitivity in the best model for cropped images with gross diagnoses. This class seems to be the only one that had great improvement when cropping was applied to the images.

Besides gross diagnosis, samples submitted to histopathology were given an overall histopathology diagnosis. This process was carried out to obtain a confirmatory diagnosis, especially for the AIP class, since histopathology findings are considered the “gold standard” [18,27,28]. When unaltered images and histopathology diagnoses were used, BIP lung images were 100% correctly classified. However, the model performed poorly on specificity due to an increase in false positives derived from BRD and AIP image misclassifications. Unfortunately, the use of histopathology labels did not improve the sensitivity of the AIP lung images in either one of the classification models. The gross appearance of the lungs used by the image classification models can prevent the correct histopathological label being assigned to the image. Variability in gross lesions of AIP encompass abnormalities affecting mostly the caudodorsal lungs or the lung in its entirety [16,17,18]. In addition, interstitial emphysema and emphysematous bullae can be present but are not pathognomonic of AIP. Because the presence of interstitial emphysema and emphysematous bullae are not particularly indicative of AIP, it is possible that a gross diagnosis of AIP will not be confirmed histopathologically even though it received an AIP gross diagnosis. Previous studies reported that only 67% of AIPs diagnosed grossly were confirmed histologically [17]. This suggests again that what is visible to the eyes and to the image classification models might not be sufficient to achieve high sensitivity and to correctly classify by image alone. It is possible that usage of samples collected only from the right side of the lung could improve the classification accuracy. However, the goal was to test if only the right-side lung images would correctly classify the overall histopathology diagnosis representing the entire lung.

The differences in lung lesion patterns presented in the images were most likely not unique and specific enough to allow the convolutional neuro networks to differentiate AIP only and BRD only from BIP lesions. Similar constraints were found in microscopy images containing high densities of cells (e.g., microglia and neurons) used in automated image acquisition [29]. To work around the constraints of different contrasts, cell types, and probe differences, pattern recognition has been proposed to find image sections that can distinguish between groups. Once this is complete, images sections can be fed into an image classification model [30]. This approach may be an option to refine image classification models in terms of classifying sections of the lung images as AIP, BIP, BRD, or undifferentiated. Nevertheless, microscopic images are often taken using similar equipment with low variation compared to images taken at field trials.

Cropping images to improve classification models by reducing the background noise did not greatly improve classification outcomes. Images on this project were collected during field necropsies; it was not uncommon that images presented variation in light exposure, image shadows, and inconsistent distance between the camera and the animal carcass. These challenges have been previously discussed as a concern regarding the reliability in images collected in the field (e.g., farms) [31,32]. A study developed to predict pig body weight reported that variance in predictions occurred when the background, light, and animal color changed [33].

## 5. Conclusions

The image classification models developed in this study presented moderately high sensitivities; however, the models still need refinement to be used widely in feedlot settings. Fine-tuning appears to be necessary, particularly regarding light exposure and image shadows. Future studies should address those challenges by testing different light exposures and backgrounds. However, our goal is to produce an image classification model that can be used in the field without the need for special tools and advanced experience. The available image classification methods could be used as an auxiliary tool for veterinarians performing a diagnosis based solely on images. This classification model is the first step towards assisting veterinarians to diagnose lung diseases in field necropsies using advanced technology and automated machine learning.

## Figures and Tables

**Figure 1 vetsci-10-00113-f001:**
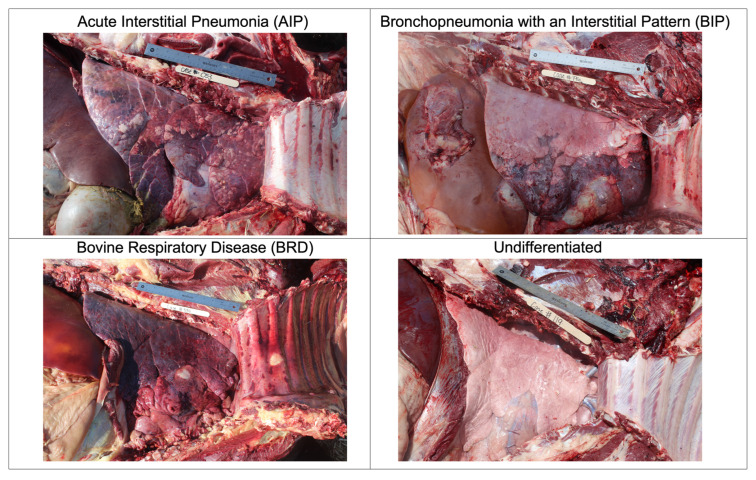
Image examples of the four possible gross diagnoses including acute interstitial pneumonia (AIP); bronchopneumonia with an interstitial pattern (BIP); bovine respiratory disease (BRD); and other lung diagnoses that were not AIP, BIP, or BRD (undifferentiated, i.e., embolic pneumonia, healthy lungs, and other diagnoses).

**Figure 2 vetsci-10-00113-f002:**
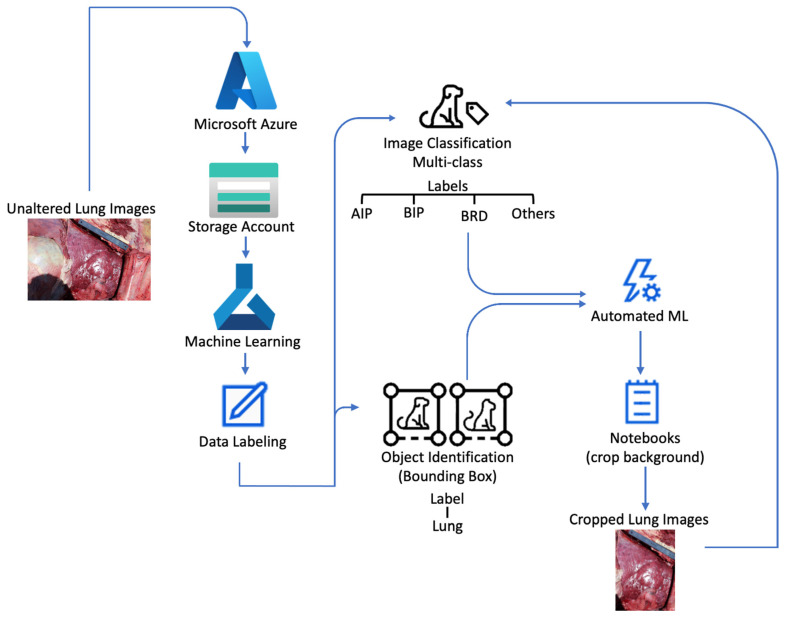
Flow chart depicting intake of unaltered lung images by Microsoft Azure into the storage account. Then, unaltered images were used to perform data labeling within the machine learning studio. Unaltered lung images received labels of acute interstitial pattern (AIP), bronchopneumonia with an interstitial pattern (BIP), bovine respiratory disease (BRD), and other lung diagnoses that were not AIP, BIP, or BRD (undifferentiated). Data tables containing the image and label were exported and fed to the automated ML. In addition, unaltered lung images were labeled using object identification (bounding box) over the lung location. Data tables containing images and binding boxes were fed to automated ML to predict lung location. Then, images were cropped using a notebook code and fed back to image classification to follow the same process as the unaltered lung images.

**Figure 3 vetsci-10-00113-f003:**
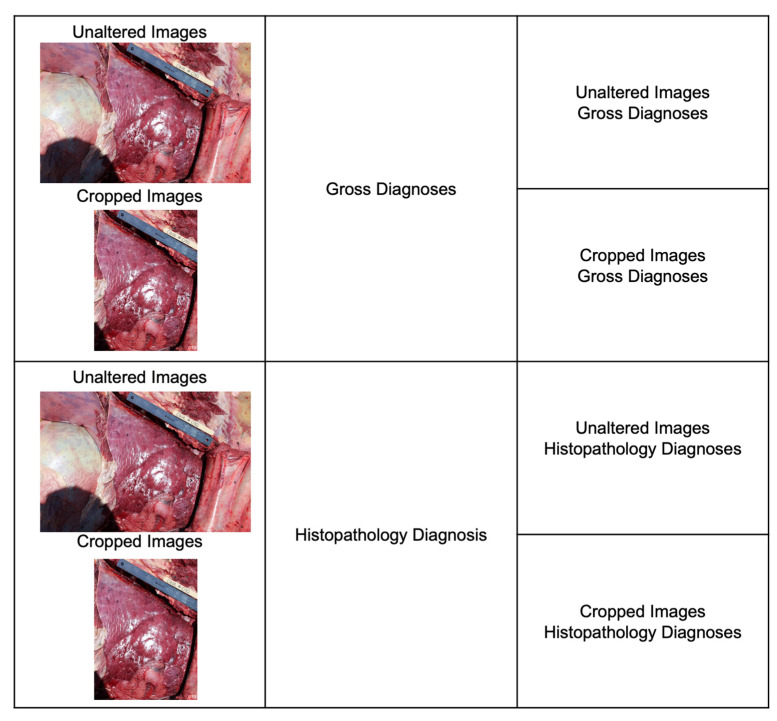
Four image classification data sets were created pairing unaltered and cropped lung images to their respective gross and histopathological diagnoses.

**Figure 4 vetsci-10-00113-f004:**
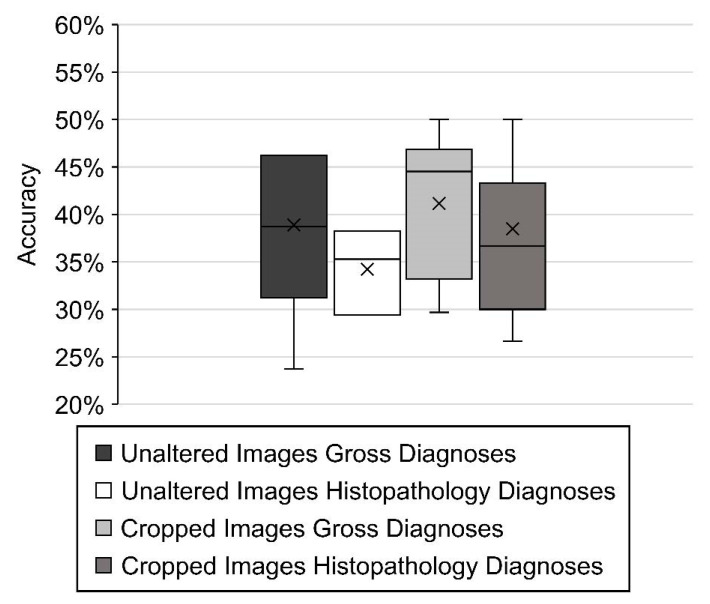
Box plot of image classification best model accuracies using the automated ML feature of Microsoft Azure for right lateral lung cattle images using the following data sets: unaltered images and their respective gross diagnoses, unaltered images and their respective histology diagnoses, cropped images and their respective gross diagnoses, and cropped images and their histology diagnosis. The variation is the representation of the epoch level within training. The × within box represents the mean, while the line represents the average.

**Table 1 vetsci-10-00113-t001:** Descriptive table of label distribution per group of images used to create image classification models, from deceased feedlot cattle right lateral lung images.

Dataset	*n* ^1^	BIP ^2^	AIP ^3^	BRD ^4^	Undifferentiated ^5^
Unaltered images gross diagnoses	398	141	44	148	65
Cropped images gross diagnoses	318	120	41	113	44
Unaltered images histopathology diagnoses	167	67	21	57	22
Cropped images histopathology diagnosis	149	61	16	55	17

^1^ Total number of images labeled; ^2^ bronchopneumonia with an interstitial pattern; ^3^ acute interstitial pneumonia; ^4^ bovine respiratory disease; ^5^ any other lung image not presenting BIP, AIP, or BRD.

**Table 2 vetsci-10-00113-t002:** Performance estimates resulting from the image classification models using the automated ML feature of Microsoft Azure for right lateral lung cattle images using the following data sets: unaltered images and their respective gross diagnoses, unaltered images and their respective histology diagnoses, cropped images and their respective gross diagnoses, and cropped images and their histology diagnosis.

Class	*n*	Se% ^5^	Sp% ^6^	PPV% ^7^	NPV ^8^	AUC ^9^	Accuracy
*Unaltered images gross diagnoses*							
AIP ^1^	14	14%	98%	67%	85%	0.68	83%
BIP ^2^	23	65%	68%	45%	82%	0.7	67%
BRD ^3^	29	69%	52%	45%	75%	0.65	58%
Undifferentiated ^4^	14	0%	100%	0%	82%	0.79	82%
*Cropped images gross diagnoses*							
AIP	8	0%	100%	0%	76%	0.44	76%
BIP	10	100%	13%	32%	100%	0.61	38%
BRD	10	20%	100%	100%	75%	0.71	76%
Undifferentiated	6	17%	100%	100%	84%	0.76	85%
*Unaltered images histopathology diagnoses*							
AIP	3	23%	98%	75%	83%	0.46	83%
BIP	30	72%	61%	50%	81%	0.71	65%
BRD	23	31%	80%	40%	73%	0.63	65%
Undifferentiated	8	70%	89%	54%	94%	0.91	86%
*Cropped images histopathology diagnosis*							
AIP	3	0%	96%	0%	89%	0.66	86%
BIP	10	60%	60%	42%	75%	0.49	60%
BRD	15	60%	73%	69%	64%	0.58	66%
Undifferentiated	2	0%	92%	0%	92%	0.51	86%

^1^ AIP—acute interstitial pneumonia; ^2^ BIP—bronchopneumonia with an interstitial pattern; ^3^ BRD—bovine respiratory disease; ^4^ undifferentiated—other lung diagnoses that were not AIP, BIP, or BRD; ^5^ Se—sensitivity; ^6^ Sp—specificity; ^7^ PPV—positive predictive value; ^8^ NPV—negative predictive value; ^9^ AUC—area under the curve.

## Data Availability

Not applicable.

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
