# Peer review of "Image Classification and Automated Machine Learning to Classify Lung Pathologies in Deceased Feedlot Cattle"

_vetsci, 2023, doi:10.3390/vetsci10020113_

Round 1

Reviewer 1 Report

Postmortem examination provides key information to feedlot operations, especially for BRD diagnosis. Unfortunately, interpretation of lung lesions can suffer from inaccuracy and lack of reproducibility, which could be overcome by using images and machine learning techniques. This study aimed at (i) developing image classification models for images of right lateral opened chests of feedlot cattle and at (ii) determining accuracies of these models for diagnosing BIP, AIP, BRD and undifferentiated lung lesions. 

This manuscript is well written and provides new and interesting information.

Please find below some general and specific comments to consider.

Line 105: please define the type of cattle necropsied (i.e. average body weight, days on feed, breed). How many veterinarians scored the lungs? Only one? Who took the pictures (the veterinarian or the technicians)? Some information on the quality of the camera used to take pictures would be of interest for the reader.

Line 116: Four lung samples were used for histology including the left lung lobes. Would it be possible to take only the histological results of the right lung lobes (as only the right lateral lung images were collected and analyzed). The fact that histopathological results from the left lung lobes were used for histopathological diagnosis could explain a low agreement between histopathological diagnosis and image classification. This point could be discussed.

Line 140: "Then, use these images" => please add the subject of the sentence (unclear as written).

Line 251: please replace "bronco-interstitial" by broncho-interstitial.

Line 298: "The gross appearance of the lungs used by the image classification models can mislead the correct histopathological label assigned to the image" Please cite the paper of Haydock et al., which recently showed that "Gross diagnosis of BIP had 83% sensitivity and 73% specificity relative to histopathology" Haydock LAJ, Fenton RK, Sergejewich L, Veldhuizen RAW, Smerek D, Ojkic D, Caswell JL. Vet Pathol. 2023 Jan 10:3009858221146092. doi: 10.1177/03009858221146092

Author Response

Dear Reviewer 1,

First, we would like to thank you for taking the time to review our paper and make good suggestions to improve our manuscript. Below are some of your concerns and suggestions and our respective answers:

Q: Line 105: please define the type of cattle necropsied (i.e. average body weight, days on feed, breed). How many veterinarians scored the lungs? Only one? Who took the pictures (the veterinarian or the technicians)? Some information on the quality of the camera used to take pictures would be of interest for the reader.

A: Camera specifications and number of veterinary technicians and veterinarian were added to line 105: “Necropsies were done by a team of 4 veterinary technicians with the supervision of a veterinarian. A total of 398 images were recorded, using a Canon EOS 4000D DSLR Camera with EF-S 18-55mm lens (Canon Inc., Tokyo, Japan), and uploaded to a storage account container within the Microsoft Azure platform.”

The definition of cattle necropsied was added to the results section, line 176 “Images taken encompass a cohort of 398 necropsied cattle with estimated weight at time of death of 445  93 kg and 90  52 days on feed, representing mainly beef-cross breeds (n = 372), beef-dairy-cross breeds (n = 23), and dairy breeds (n = 3).”

Regarding the number of veterinarians that scored the lungs, clarification is in line 114 “The necropsy technicians and a veterinarian conjointly defined the gross diagnoses for each lung image during the necropsy procedure.”

Q: Line 116: Four lung samples were used for histology including the left lung lobes. Would it be possible to take only the histological results of the right lung lobes (as only the right lateral lung images were collected and analyzed). The fact that histopathological results from the left lung lobes were used for histopathological diagnosis could explain a low agreement between histopathological diagnosis and image classification. This point could be discussed.

A: In our image classification model we tested if only using the right-side lung picture was enough to classify lung pathologies affecting the entire lung. Our hypothesis was that the computer might identify things not seen grossly and that is why all four lung samples were used to produce the overall histopathology diagnoses. Some clarifications were added to the manuscript.

Line 121: “The sample collection for histopathology was developed with the intent to capture major lung sections commonly affected by pulmonary pathologies (cranioventral and caldodorsal lobes).”

Line 333: “It is possible that usage of samples collected only from the right side of the lung could improve the classification accuracy. However, the goal was to test if only the right-side lung images would correctly classify the overall histopathology diagnosis representing the entire lung.”

Q: Line 140: "Then, use these images" => please add the subject of the sentence (unclear as written).

A: Thank you for pointing that out. A subject was added to the sentence. “Then, unaltered images were used to perform data labeling within the machine learning studio.”

Q: Line 298: "The gross appearance of the lungs used by the image classification models can mislead the correct histopathological label assigned to the image" Please cite the paper of Haydock et al., which recently showed that "Gross diagnosis of BIP had 83% sensitivity and 73% specificity relative to histopathology" Haydock LAJ, Fenton RK, Sergejewich L, Veldhuizen RAW, Smerek D, Ojkic D, Caswell JL. Vet Pathol. 2023 Jan 10:3009858221146092. doi: 10.1177/03009858221146092

A: Thank you for the suggestion. We made adjustments to the Haydock citations throughout the manuscript. Their two papers were published after we made the submission, and we are glad we can reference them now.

Reviewer 2 Report

In this study Bortoluzzi et al reported that they have developed image classification models to assist veterinarian’ lung diseases diagnostic in field necropsies. Artificial intelligence has been applied to image analysis for the diagnostic method. As to the study, I have several concerns.

1.     The aim of this study is to establish a technique to differentiate Bovine Respiratory Disease (BRD), Acute Interstitial Pneumonia (AIP), Bronchopneumonia with an interstitial pattern (BIP), and other lung diagnoses that were not belonging to BRD, AIP, and BIP (Undifferentiated). The authors should show typical images of each type. This is not difficult for the authors because they have large amounts of raw images from field necropsy.

2.     Generally, the typical lung lesions are not evenly and entirely distributed in lung. Particularly, some lesion foci are visible in part of the lung and the remaining part are looks healthy. In this report the authors run analysis of four lung samples, including left cranioventral lobe; right cranioventral lobe; left caudal dorsal lobe; right caudal dorsal lobe. I wonder if the lesions are not located at these samples, the results what they got will be still accurate?

Author Response

Dear Reviewer 2,

We would like to thank you for reviewing our manuscript and making significant contributions to improve it. Below you will find your questions and suggestions and their respective answers.

Q: The aim of this study is to establish a technique to differentiate Bovine Respiratory Disease (BRD), Acute Interstitial Pneumonia (AIP), Bronchopneumonia with an interstitial pattern (BIP), and other lung diagnoses that were not belonging to BRD, AIP, and BIP (Undifferentiated). The authors should show typical images of each type. This is not difficult for the authors because they have large amounts of raw images from field necropsy.

A: Image examples were added as Figure 1. Thank you for the suggestion.

Q: Generally, the typical lung lesions are not evenly and entirely distributed in the lung. Particularly, some lesion foci are visible in part of the lung and the remaining part are looks healthy. In this report the authors run analysis of four lung samples, including left cranioventral lobe; right cranioventral lobe; left caudal dorsal lobe; right caudal dorsal lobe. I wonder if the lesions are not located at these samples, the results what they got will be still accurate?

A: The sample collection areas are the main sections in the lungs affected by pulmonary disease, and therefore they were the selection sites. In the collected cases, we aimed to represent the border between gross affected areas (areas with lesion) and gross healthy areas (areas without lesion) in the same sample, ensuring that affected areas were always being collected. However, it is possible that in some cases the lesions were missed resulting in an inaccurate histopathological diagnosis. The authors believe that this represents variability encountered in lung diagnostics.

Reviewer 3 Report

The manuscript deals with an absolutely interesting topic to favor diagnostic interpretation by providing fixed and standardized criteria. However, the absence of histopathological images associated with macroscopic images makes the work incomplete, in my opinion, since they are called into question in the text. I am sure that perfecting the whole method by associating, also for diagnostic evidence, histopathology will certainly contribute to a greater validity of the work.

Author Response

Q: The manuscript deals with an absolutely interesting topic to favor diagnostic interpretation by providing fixed and standardized criteria. However, the absence of histopathological images associated with macroscopic images makes the work incomplete, in my opinion, since they are called into question in the text. I am sure that perfecting the whole method by associating, also for diagnostic evidence, histopathology will certainly contribute to a greater validity of the work.

A: Thank you for your suggestion; If we understood correctly, your suggestion was to use both gross images (right-side lung images) and histopathological images to classify lung syndromes. However, the aim of the study was to develop image classification models using machine learning to determine respiratory syndromes diagnostic accuracy in right-lateral necropsied feedlot cattle lung images in the field. These lung images can be easily collected during a field necropsy and tested using the image classification developed. The use of histopathological images to classify lung pathologies was not the objective of our study, mainly because they are not readily and timely available for veterinarians at the time of the field necropsy. In addition, not all cattle necropsied in feedlots have their samples submitted to histopathological analysis. Our goal in using the histopathological diagnoses was to improve our image classification, especially in cases where it is considered the “gold standard” (e.g., AIP). However, it would be interesting to create another project to classify histopathological images that could aid clinical pathologists in their diagnoses.

Round 2

Reviewer 2 Report

Since my concerns have been addressed, I agree to be accepted for publication.

Reviewer 3 Report

The manuscript appears more detailed in some sections. Once again, in my opinion, the absence of histopatological patterns slightly penalizes the paper. I would also like to suggest to replicate the study on the other side, if working conditions allow it.